# Online Sequential Learning from Physiological Data with Weighted Prototypes: Tackling Cross-Subject Variability

## Abstract

Online Continual Learning (OCL) enables machine learning models to adapt to sequential data streams in real time, especially when only a small amount of data is available. However, applying OCL to physiological data such as electroencephalography (EEG) and electrocardiography (ECG) is often complicated by inter-subject variability, which can lead to catastrophic forgetting and performance degradation. Existing OCL methods are currently unable to effectively address this challenge, leading to difficulties in retaining previously learned knowledge while adapting to new data. This paper presents Online Prototype Weighted Aggregation (OPWA), a novel method specifically designed to address the problem of catastrophic forgetting in the presence of inter-subject variability through the use of prototypical networks. OPWA facilitates the retention of knowledge from past subjects while adapting to new data streams. The OPWA method uses an innovative prototype aggregation mechanism that fuses intra-class prototypes into generalized representations by accounting for both within-class and inter-class variation between subjects. Extensive experiments show that OPWA consistently outperforms existing OCL methods in terms of fast adaptation and mitigation of catastrophic forgetting on different physiological datasets with different modalities, and provides a robust solution for learning on sequential data streams.

## 1 Introduction

Intelligent machines equipped with artificial intelligence (AI) are increasingly becoming indispensable partners for humans in various sensitive and time-critical domains, ranging from search and rescue missions to space exploration Layton (2021); El Alami et al. (2023). The dynamics of real-world environments present complex tasks that can significantly increase the human workload and impair decision-making capabilities. To optimize human-machine collaboration Shively et al. (2018), AI systems need to understand human preferences so that they can adapt their behaviour to the mental workload of their human counterparts. Non-invasive technologies Gu et al. (2021), such as wearable devices, offer a promising approach to collect implicit feedback with minimal distraction.

Non-invasive technologies capture physiological signals such as electroencephalography (EEG) and electrocardiography (ECG), which offer valuable insights into human mental workload and enable real-time monitoring of stress levels. However, classical machine learning (ML) requires large, annotated datasets for effective stress prediction, which are often not available in the real world for various reasons. For instance, publicly available datasets are usually collected in specific environments using specialized emotion elicitation techniques which often cannot be generalized to all situations Khare et al. (2024). In addition, these datasets usually represent a limited number of classes Duan et al. (2024b). Dynamic and time-critical environments lead to different levels of stress in individuals, which can vary considerably from one environment to another. This limitation makes it necessary to train models online on continuous data streams, a process known as Online Continual Learning (OCL). Unlike continual learning, which requires multiple passes over data, OCL processes data incrementally in real-time, observing each sample or mini-batch only once to adapt to dynamic scenarios Soutif-Cormerais et al. (2023). Data arrives incrementally in OCL, which can be either domain incremental (where the input distribution changes over time) or class incremental (where new classes are introduced over time). Existing applications of physiological signals can

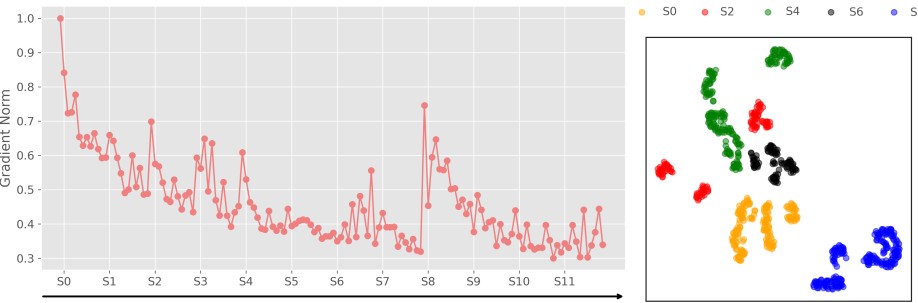

Figure 1: **Right Panel:** Gradient norm during online learning of a four-class Valence and Arousal classification model on subject-wise sequential EEG data from the AMIGOS dataset, illustrating the effects of cross-subject variability. **Left Panel:** UMAP visualization of the embedding space from the learned feature extractor, with different colours representing embedding vectors from various subjects from a particular class.

benefit from OCL and enable systems to seamlessly adapt to dynamic scenarios, such as air traffic management Aricó et al. (2016b;a), aircraft operations Dehais et al. (2019); Almogbel et al. (2019); Lim et al. (2017), entertainment Hafeez et al. (2021) and search and rescue operations Lim (2021).

Physiological data show high inter-subject variability in the representation of generic emotions, as the data are influenced by factors such as biological differences, contextual factors and personal experiences. This variability makes it difficult to develop a generalizable model applicable to all subjects Choi et al. (2020). Additionally, the nature of OCL further complicates these challenges, as the data is continuously streaming, resulting in a lack of comprehensive statistical information about entire training data. Figure 1 illustrates the effects of inter-subject variability on online learning of convolutional neural network (CNN) in the OCL paradigm with EEG data from the AMIGOS database. The left panel shows the trajectory of the gradient norm during sequential learning across subjects. As training progresses from subject $s0$ to $s1$, the gradient norm gradually decreases; however, it increases at $s2$, which indicates a shift in the subject. This pattern becomes even more pronounced at $s8$, highlighting significant inter-subject variability that can lead to catastrophic forgetting, where the model can lose its prior knowledge due to large gradient descent steps on the current data. The right panel in Figure 1 visualizes this subject shift variability using UMAP [1]. This shows the embedding vectors for different subjects for a particular class, highlighting the unique data distribution of $s8$ that contribute to catastrophic forgetting in OCL settings.

In this work, we propose a robust OCL approach to mitigate the problem of catastrophic forgetting in the presence of inter-subject variability. Inspired by prototypical networks Snell et al. (2017), we develop Online Prototype Weighted Aggregation (OPWA), a novel approach to ensure the retention of previous knowledge while adapting the model to the new subjects. Existing approaches Wei et al. (2023); De Lange & Tuytelaars (2021) that incorporate prototypical loss in OCL use momentum-based strategies to update class prototypes based on the mean embeddings of the current data stream. However, such naive prototype updates often result in catastrophic forgetting, as inter-subject variability in OCL causes prototypes to drift away from representations of past data. To address this, our approach introduces intra-class prototype weighting, a key innovation that goes beyond traditional momentum-based updates. While methods like CoPE De Lange & Tuytelaars (2021) incrementally adapt prototypes to new data streams, they fail to account for distributional shifts across subjects, increasing the risk of forgetting. In contrast, our method employs a dynamic weighted aggregation scheme that adapts to these shifts, ensuring more robust and stable prototype representations. This dynamic weighting, determined by the distances between prototypes, prioritizes prototypes that are more stable and reflect the true data distribution. To the best of our knowledge, this weighted aggregation strategy has not been previously employed in the context of OCL. Our contributions are summarized as follows: **(1)** We propose OPWA to address the challenge of inter-subject variability in OCL, introducing a robust prototype aggregation mechanism that synthesizes global prototypes from cross-subject prototypes. These global prototypes act as generalized class anchors that serve as the centroids of embedding vectors and improve the prototypical loss for OCL.**(2)** Our approach

---

[1]https://umap-learn.readthedocs.io/en/latest/

considers the underlying data distribution of each subject while constructing global prototypes. The aggregation mechanism takes into account intra-class variance and thereby naturally improves the decision boundaries between inter-class prototypes. **(3)** We demonstrate the effectiveness of the proposed method through extensive experiments with various physiological datasets, highlighting its superior performance in both adaptation and forgetting mitigation across different modalities.

## 2 RELATED WORK

Continual learning frameworks are designed to learn from data streams that may undergo significant distribution shifts. Recently, several approaches have been introduced to tackle catastrophic forgetting caused by these shifts, which can be categorized into three main types.

Regularization-based approaches apply regularization terms to control the process of parameter updates and can be further categorized into weight regularization Kirkpatrick et al. (2016); Ritter et al. (2018); Schwarz et al. (2018) and function regularization Dhar et al. (2018); Hung et al. (2019); Qin et al. (2021); Miao et al. (2022). The weight regularization selectively controls the changes in network parameters. Alternatively, function regularization approaches focus on the intermediate or final outputs of the prediction function. Replay-based methods have also been proposed that store a limited number of samples from previously observed distributions and use them for KD or joint training with current data. Replay-based methods include GEM Lopez-Paz & Ranzato (2017), iCARL Rebuffi et al. (2017), reservoir sampling Vitter (1985) and variants of reservoir sampling Aljundi et al. (2019a;b). Most of recent works in OCL have focused on computer vision (CV) tasks, particularly on image classification. However, when applied to physiological datasets, these approaches may not perform well, as these datasets present unique challenges Nakisa et al. (2018). For example, OnPro Wei et al. (2023) introduced prototype equilibrium to prevent shortcut learning, a common problem in image data where models can learn background features to discriminate between classes.

Very few studies specifically address the classification of physiological data Duan et al. (2024b;a). A recent study Duan et al. (2024b) proposed AMBM, a meta-learning approach to tackle catastrophic forgetting and facilitate rapid adaptation in the presence of subject shifts in EEG signals. This method is based on the Model-Agnostic Meta-Learning (MAML) Finn et al. (2017). In the base loop, a contrastive loss Khosla et al. (2020) is applied to the current data streams for fast adaptation. During the meta-phase, memory replay using reservoir sampling Vitter (1985) is used for rehearsal, with an adaptive learning rate to mitigate forgetting. However, a significant drawback of this method is the overfitting of the current data stream, as the contrastive loss is explicitly applied to the current data stream for several steps in the inner loop, leading to forgetting of the knowledge gained on previous subjects. In addition, AMBM is computationally intensive due to its bilevel optimization that separates adaptation and generalization processes, resulting in higher computational demands. In contrast, our approach efficiently preserves past knowledge while adapting to the new data stream, treating both objectives together in a single process. We incorporate a prototypical loss that utilizes enhanced prototypes derived from an effective prototypes aggregation mechanism. Another work Duan et al. (2024a) proposes to maintain a balanced representation across subjects by considering data volume and informativeness, using clustering to detect subject shifts, and selectively replacing less informative or overrepresented samples in the memory buffer. However, the use of gradient norms to measure informativeness may be less effective in dealing with noisy or non-stationary data, which could lead to suboptimal memory representation in highly dynamic scenarios.

Elastic Weight Consolidation (EWC) Kirkpatrick et al. (2016) is a domain adaptation technique that helps prevent catastrophic forgetting by penalizing changes to important parameters using the Fisher Information Matrix (FIM). However, its effectiveness depends on the amount of data, as the Fisher information requires a sufficient amount of data for accurate estimation, and it incurs high computational costs as the FIM needs to be updated frequently, which makes it impractical in OCL settings. CLUDA Ozyurt et al. (2022) is a contrastive learning based domain adaptation approach designed for time series data which aligns contextual representations between source and target domains through adversarial training and contrastive learning. Although it is effective for domain adaptation tasks, its application in OCL is challenging because there is no static source domain. Although the memory buffer could serve as a pseudo-source domain, its dynamic and limited nature makes it difficult to achieve robust generalization and retain knowledge across sequential subjects.

The most relevant works to ours are OnPro Wei et al. (2023) and CoPE De Lange & Tuytelaars (2021), as they also use prototypes, but in the context of CIL. OnPro introduced prototype equilibrium to prevent shortcut learning in image data by applying contrastive loss between the prototypes of the original data and those of the augmented views. On the other hand, CoPE employs prototype evolution, which allows the prototypes to evolve naturally using a momentum-based approach but does not consider subject shifts, leaving a potential bias towards new subjects. In contrast, our method specifically addresses subject shifts by deriving generalized prototypes that represent the entire data distribution, ensuring generalizability in the presence of cross-subject variability.

## 3 METHOD

In this section, we present OPWA, a novel approach that effectively addresses the challenge of inter-subject variability in OCL. The core strength of our method lies in the integration of prototype aggregation with normalized weights, ensuring that subject-specific prototypes contribute appropriately to the overall class representation without being biased by any single prototype. While momentum-based methods, such as CoPE De Lange & Tuytelaars (2021), gradually update prototypes to accommodate new data streams, they often drift away from representations of past subjects over time, increasing the risk of forgetting. In contrast, our approach incorporates a weighted aggregation scheme that adapts to distributional shifts between subjects and ensures a more robust representation of class prototypes. This dynamic weighting, determined by the distances between prototypes, allows us to prioritize those that are more stable and reflect the true data distribution. OPWA proposed method provides a refined approach for managing prototypes, especially in scenarios with significant subject shifts, and represents a substantial advancement over existing techniques.

### 3.1 PROBLEM FORMULATION

Let $\mathcal{S} = \{\mathcal{S}_1, \mathcal{S}_2, ..., \mathcal{S}_s\}$ represent the sequence of subjects arriving sequentially. Each subject $s$ generates a labeled data $D_i^s = \{(\boldsymbol{x}_j, y_j)\}_{j=1}^{B}$ comprising $B$ labeled data points. Here $\boldsymbol{x}_j$ denotes the physiological signal segment and $y_j$ is the corresponding label. We utilize a memory replay buffer, denoted as $\mathcal{M}$. This memory bank stores a subset of past data points, allowing the model to revisit and enforce learning of past subjects. At each time step $t$, the model receives a mini-batch of training data $\mathcal{D} = \{\mathcal{D}_s^t, \mathcal{D}_b\}$, where $\mathcal{D}_s^t$ are the data from the subject $s$ at time step $t$, and $\mathcal{D}_b$ includes samples drawn from the memory buffer $\mathcal{M}$. The model consists of three key components: an encoder network $f$, a projection head $g$, and a classifier $\phi$. Each sample $x$ in the incoming data $\mathcal{D}$ is transformed into a projected vectorial embedding $\boldsymbol{z}$ through the and encoder and projector:

$$\boldsymbol{z} = g(f(\boldsymbol{x}; \theta_f); \theta_g), \tag{1}$$

where $\theta_f$ and $\theta_g$ represent the parameters of $f$ and $g$, respectively. At each time step $t$, the online prototype for each class $k$ is calculated as the mean representation in the mini-batch:

$$\boldsymbol{p}_i = \frac{1}{|\boldsymbol{z}_k|} \sum_j \boldsymbol{z}_j . \mathbf{1}_{y_j=k} \tag{2}$$

where $|\boldsymbol{z}_k|$ denotes the number of embeddings belonging to class $k$ in the mini-batch, and $\mathbf{1}$ is the indicator funtion. This process results in a set of $K$ online prototypes derived from data $\mathcal{D}$, $\mathcal{P} = \{\boldsymbol{p}_k\}_{k=1}^K$. The prototypical loss function is defined as follows:

$$\mathcal{L}_P = -\frac{1}{|\boldsymbol{z}|} \sum \log \left( \frac{\exp\left(-\|\boldsymbol{z}_j - \boldsymbol{p}_{y_j}\|^2\right)}{\sum_k \exp\left(-\|\boldsymbol{z}_j - \boldsymbol{p}_k\|^2\right)} \right) \tag{3}$$

where $\boldsymbol{z}_j$ denotes the embedding vector of the sample $\boldsymbol{x}_j$, and the index $k$ runs over all classes. This loss function is prototypical loss Laenen & Bertinetto (2020) that encourages the model to minimize the distance between each embedding vector and its corresponding prototype.

### 3.2 PROTOTYPES EVOLUTION

As training progresses through successive mini-batches, the prototypes should be continuously refined to better represent the centroids of embedding vectors observed so far. This is typically

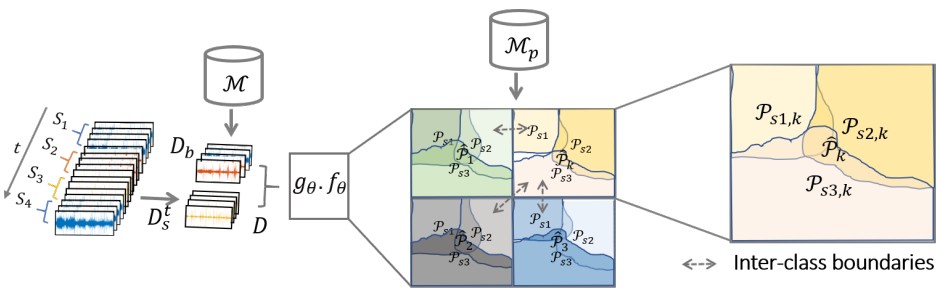

Figure 2: OPWA Framework: The feature extractor $f_\theta$ and projection head $g_\theta$ map inputs $\mathcal{D}$ into the embedding space. $\mathcal{P}_{s1,k}$ represents the class $k$ prototype for subject $s1$, while $\hat{\mathcal{P}}_k$ is the generalized prototype, formed by aggregating prototypes from subjects $s1$, $s2$, and $s3$.

achieved using a momentum-based approach De Lange & Tuytelaars (2021), which gradually updates each prototype $\bar{p}_k$ after each mini-batch:

$$\bar{p}_k^{t+1} \longleftarrow \alpha \bar{p}_k^t + (1 - \alpha)p_k^{t+1}, \tag{4}$$

where $\bar{p}_k^t$ is the prototype for class $k$ at time step $t$, $p_k^{t+1}$ is the prototype computed from current mini-batch at time step $t+1$ and $\alpha \in [0, 1]$ controls the balance between historical and new information. The momentum-based update strategy assumes relative stability of the data distribution over time. Although prototypes evolve through a momentum-based approach ensures smooth and stable updates, this method can fall short during sequential learning on subjects with high inter-subject variability, Fig. 1. During sequential learning, subject changes introduce significant variations in physiological responses, leading to substantial changes in the underlying data distributions. As a result, the prototypes evolution approach may cause prototypes to drift away from global representations. This drift can greatly influence prototypes, possibly causing them to adapt primarily to new data and disregard previous subjects which, in turn, can lead to increased risk of catastrophic forgetting. To overcome these challenges, our approach introduces the prototype aggregation mechanism that dynamically adapts the prototypes while ensuring robust learning across subjects.

### 3.3 PROTOTYPES WEIGHTED AGGREGATION

Figure 2 illustrates the proposed framework. As mentioned above, developing prototypes with a naïve strategy that neglects subject shifts can lead to catastrophic forgetting. Even a single outlier with a unique data distribution can significantly degrade the performance of a previously trained model. To address this problem, we propose OPWA that accounts for cross-subject variability and aggregates intra-class prototypes of seen subjects based on their relative importance for generalization. This approach aims to fuse within-class prototypes into generalized representative prototype that help retain previous knowledge while adapting to current data.

Consider that the prototype $\bar{p}_{k,s}$ is evolved during online learning at time step $t + n$ using data from subject $s$. When a subject shift occurs from $s$ to $s + 1$, we store all prototypes $\mathcal{P}_s = \{\bar{p}_{k,s}\}_{k=1}^K$ corresponding to subject $s$ in a memory buffer $\mathcal{M}_p$. Before proceeding with training on subject $s + 1$, our goal is to aggregate the intra-class prototypes of all subjects previously stored in $\mathcal{M}_p$ into a generalized prototype given by:

$$\hat{p}_k = \sum_{s \in \mathcal{S}^t} w_{k,s} \bar{p}_{k,s} \tag{5}$$

Equation 5 performs a weighted aggregation of the prototypes from all seen subjects $\mathcal{S}^t$ at time step $t$, resulting in a global representation prototype for class $k$, denoted as $\hat{p}_k$. This leads to a global prototype set $\hat{\mathcal{P}} = \{\hat{p}_k\}_{k=1}^K$, which consists of the global prototypes for all $K$ classes. In Figure 2, note that $\hat{p}_k$ represents the generalized centroids of the embeeding vectors for all seen subjects. Incorporating these prototypes into the prototypical loss in Equation 3 would effectively address catastrophic forgetting. However, their effectiveness depends on their contribution weights $w_{k,s}$ used during the aggregation process. We consider Intra-class variances between prototypes stored in $\mathcal{M}_p$ to determine their contribution toward generalization. These variances reflect the deviation of each subject's data distribution and serve as a metric to determine the impact on generalization.

While weighted prototype aggregation leads to generalized prototypes, the approach naturally enhances the boundaries between the inter-class global prototypes.

**Intra-Class Weights:** As illustrated in Figure 2, the feature extractor projects mini-batches of training data $\mathcal{D}$ into the embedding space. For the sake of clarity, we will focus on the embeddings of a single class within the embedding space. The embedding vectors of each subject occupy a specific region in the embedding space that determines its variability relative to the other subjects. In an ideal scenario with no variability, these regions would completely overlap, and the centroid of each class would accurately represent the true generalized prototype of that class. However, the distributional shifts between subjects result in their samples being projected into distinct, personalized regions within the embedding space. Thus, existing prototype evolution strategies, such as momentum-based approaches, are vulnerable to distribution shifts, favoring the most recent subjects, and leaving potential biases within the prototypes. To overcome this bias, we fuse prototypes using a weighted aggregation approach that considers the importance of each personalized prototype based on its generalization ability. As illustrated in Figure 2, a smaller distance between intra-class prototypes indicates less distributional shift in the respective data, making them more representative of the overall population. On the other hand, a prototype that is significantly further away indicates a larger distribution shift and should be given less priority to avoid bias in centroid. To this end, we compute the pairwise Euclidean distances between intra-class prototypes and employ a Gaussian kernel to transform these distances into weights.

$$w_{ij} = \exp\left(-\frac{\|\bar{\boldsymbol{p}}_i - \bar{\boldsymbol{p}}_j\|^2}{2\sigma^2}\right), \tag{6}$$

where $\bar{\boldsymbol{p}}_i$ and $\bar{\boldsymbol{p}}_j$ represent the prototypes of subjects $i$ and $j$ stored in $\mathcal{M}_p$, and $\sigma$ is the smoothing parameter that regulates these weights. $w_{i,j}$ is the entry in the weight matrix $\boldsymbol{W}_k$ that represents the distance between prototypes $\bar{\boldsymbol{p}}_i$ and $\bar{\boldsymbol{p}}_j$.

**Overall Framework:** We incorporate normalized weights into the prototypes aggregation as normalization is essential to avoid skewing the contributions of the prototypes and enusres that they collectively represent the contribution of each prototype relative to the others. To ensure that the weights in each row sum up to 1, we normalize each row of $\boldsymbol{W}_k$. The normalization is applied as:

$$\tilde{w}_{ij} = \frac{w_{ij}}{\sum_j w_{ij}} \tag{7}$$

where $\tilde{w}_{ij}$ represents normalized weight between prototypes $\bar{\boldsymbol{p}}_i$ and $\bar{\boldsymbol{p}}_j$ ensuring that: $\sum_j \tilde{w}_{ij} = 1$. The normalized weights are then applied to aggregate the prototypes for each class as follows:

$$\hat{\boldsymbol{p}}_k = \sum_{i=1}^{\mathcal{S}^t} \sum_{j=1}^{\mathcal{S}^t} \tilde{w}_{ij} \bar{\boldsymbol{p}}_{j,k} \tag{8}$$

$\hat{\boldsymbol{p}}_k$ represents the consensus mean prototype of class $k$, derived by combining the normalized weights $\tilde{w}_{ij}$ and the mean prototypes $\bar{\boldsymbol{p}}_{j,k}$ of all subjects in $\mathcal{S}^t$. This aggregation allows us to synthesize a set of generalized prototypes $\hat{\mathcal{P}}$ that serve as stable representations that are less susceptible to the biases introduced by cross-subject variability. Thus, the prototypical loss used in our approach is given by:

$$\mathcal{L}_{Proto} = -\frac{1}{|\boldsymbol{z}|} \sum \log\left(\frac{\exp\left(-\|\boldsymbol{z}_j - \hat{\boldsymbol{p}}_{y_j}\|^2\right)}{\sum_k \exp\left(-\|\boldsymbol{z}_j - \hat{\boldsymbol{p}}_k\|^2\right)}\right) \tag{9}$$

Note that equation 9 differs from equation 3 in that it employs the generalized prototype $\hat{\mathcal{P}}_{y_i}$ instead of relying solely on prototypes derived from each mini-batch. These generalized prototypes are designed for each class to accurately represent the true centroids of entire data distribution. As a result, the loss function $\mathcal{L}_{Proto}$ provides a robust solution for retaining previous knowledge despite inter-subject variability. The overall loss of our OPWA approach is given as:

$$\mathcal{L}_{OPWA} = \mathcal{L}_{CE} + \mathcal{L}_{Proto} + \mathcal{L}_P \tag{10}$$

where $\mathcal{L}_{CE} = CE(y, \phi(g(f(\boldsymbol{x}))))$ is the cross-entropy loss. While $\mathcal{L}_{CE}$ and $\mathcal{L}_P$ focuses on rapid adaptation to a new subject, $\mathcal{L}_{proto}$ ensures the retention of knowledge gained from subjects seen previously. A detailed summary of the method can be found in Appendix A.4.

## 4 EXPERIMENTS

**Dataset:** We used four publicly available datasets to evaluate our proposed method, across different subjects, conditions and data collection protocols. The datasets include AMIGOS Correa et al. (2017), DEAP Koelstra et al. (2012), PPB-EMO Li et al. (2022) and BCI-IV-2a Tangermann et al. (2012). While BCI-IV-2a focuses on motor imagery classification(left hand, right hand, tongue and both feet), the other datasets focus on emotion classification based on valence and arousal. These data sets capture various physiological signals from subjects while watching videos of specific duration (seconds) that are designed to elicit specific emotional responses characterized by valence (high/low) and arousal (high/low). Each data set consists of different channels and is recorded at different sampling rates. For example, the pre-processed AMIGOS data is downsampled to 128 Hz. We create 5-second segments for each trial with overlap between consecutive segments. Data are annotated on the basis of subjects' ratings of valence and arousal on a defined scale. More details on the data sets can be found in Appendix A.1. Our evaluation considered both EEG and ECG modalities with a four-class classification task to categorize valence and arousal. For example, in the AMIGOS dataset, subjects rate their emotional state on scales of 0 to 9 for both valence and arousal. Ratings above 5.5 indicate high valence or high arousal, while ratings below 4.5 indicate low valence or low arousal. Intermediate ratings are categorized as neutral emotions. Using this methodology, we can categorize emotions into four distinct classes: High Valence Low Arousal (HVLA), High Valence High Arousal (HVHA), Low Valence High Arousal (LVHA), and Low Valence Low Arousal (LVLA). Neutral emotions are excluded from the dataset.

**Baselines:** The baseline methods included in this study fall into four categories: **(1) Offline Learning**: demonstrates the upper bound performance of the model reached through joint learning, where the data of all subjects is available simultaneously and the model can learn offline. **(2) Domain Adaptation:** We employed EWC Kirkpatrick et al. (2016) as an additional baseline. However, since EWC cannot be directly applied in an OCL setting, we adapted it by using the memory buffer as a pseudo-source domain. Specifically, the model is trained on the memory buffer before adapting to the next subject. Moreover, we incorporated CLUDA Ozyurt et al. (2022) as an additional baseline. CLUDA represents domain adaptation approach that relies on adversarial training and contrastive learning to align contextual representations across source and target domains, making it particularly effective for tasks involving temporal data. To adapt CLUDA for OCL setting, where no explicit source domain is available, we utilized the memory buffer as a pseudo-source domain. The memory buffer, populated during sequential subject learning, served as the labeled source domain for the adaptation process. **(3) Online Learning Techniques:** We consider two online learning methods, OnPro Wei et al. (2023) and CoPE De Lange & Tuytelaars (2021), both of which employ prototypical loss functions within their frameworks. Although these techniques are primarily designed for image classification tasks, we use them as comparative benchmarks to evaluate the performance of our method against existing solutions as they utilize prototypical loss functions. **(4) Bi-level Meta Learning:** We incorporate the AMBM Duan et al. (2024b), specifically tailored for the online learning of EEG data in a sequential manner. This approach leverages a bilevel optimization framework, enabling rapid adaptation to new data in the inner loop through a contrastive loss mechanism and effectively mitigates forgetting in the meta-loop. Finally, we establish a baseline, called OCL, which applies data augmentation to the mini-batch $\mathcal{D}$ and employs cross-entropy loss.

**Model architecture and settings:** A convolution neural network (CNN) is designed for the feature extractor $f$ which consists of a 1D convolution layer featuring 32 filters with a kernel size of 7 and a stride of 2, followed by batch normalization. Next, the model includes three residual blocks, each comprising two convolution layers with kernel sizes (15, 21, and 43), followed by batch normalization. A max pooling layer with a kernel size of 4 and a stride of 4 is then applied followed by an attention layer. The feature extractor concludes with three fully connected layers containing 1024, 512, and 256 units, respectively. A projection head, $g$ ,is applied with embedding dimension of 128, followed by a linear classifier $\phi$. Relu activation functions are applied for non-linearity and Softmax function is used with classifier. Following Duan et al. (2024b); Wei et al. (2023); De Lange & Tuytelaars (2021), we adopt the reservoir sampling for the data memory buffer $\mathcal{M}$, maintaining 200 balanced data segments. Prototypes for each class from all subjects are stored in the memory buffer $\mathcal{M}_p$, which dynamically increases in size during subject shifts. However, the size of $\mathcal{M}_p$ remains significantly smaller than that of $\mathcal{M}$, as it only stores 4 prototypes per subject, with each prototype having a dimensionality of 128. For details on the hyperparameter settings, please refer to A.2.

**Evaluation Metrics:** Following Duan et al. (2024b), we adopt Average Adaptation Accuracy (AAA) and Fogetting Mitigation Accuracy (FMA) as evaluation metrics. AAA evaluates the performance on the current subject's test set immediately after training on that subject, calculated as $AAA = \frac{1}{|\mathcal{S}|} \sum_{j=1}^{|\mathcal{S}|} a_j$, where $a_j$ represents the accuracy on the test set for the subject $j$. Forgetting Mitigation Accuracy measures the ability of the model to retain knowledge about all subjects after training is completed for the last subject, expressed as $FMA = \frac{1}{|\mathcal{S}|-1} \sum_{j=1}^{|\mathcal{S}|-1} m_j$, where $m_j$ indicates the model's accuracy on the test set of the subject $j$ at the end of training. The last subject is omitted as it does not experience forgetting.

## 4.1 RESULTS AND DISCUSSION

| Dataset | Method | 90% Overlap | | 75% Overlap | | 50% Overlap | |
|---|---|---|---|---|---|---|---|
| | | AAA (Mean ± Std) | FMA (Mean ± Std) | AAA (Mean ± Std) | FMA (Mean ± Std) | AAA (Mean ± Std) | FMA (Mean ± Std) |
| BCI-IV-2a | Offline | 96.24 | | 90.96 | | 68.92 | |
| | OCL | 37.16±2.07 | 39.53±3.29 | 32.58±0.57 | 34.3±5.08 | 31.56±2.12 | 34.0±3.39 |
| | EWC | **40.31±2.38** | **45.26±1.57** | **35.88±2.07** | **39.60±8.27** | **35.45±1.08** | **37.89±8.6** |
| | CLUDA | 24.38±0.54 | 25.54±0.95 | 24.98±0.79 | 24.94±0.10 | 25.08±0.11 | 25.09±0.39 |
| | AMBM | 30.13±1.18 | 30.26±3.54 | 27.82±0.83 | 27.13±1.71 | 28.75±1.66 | 29.11±3.83 |
| | CoPE | 34.37±0.62 | 35.75±4.33 | 33.39±1.34 | 35.6±4.88 | 30.33±1.76 | 33.22±7.22 |
| | OnPro | 28.62±0.8 | 30.58±2.34 | 28.43±1.14 | 28.42±3.49 | 27.0±0.95 | 26.74±2.07 |
| | OPWA | 37.35±3.05 | 41.67±5.66 | 33.63±0.64 | 37.84±3.53 | 30.92±2.16 | 35.47±4.23 |
| DEAP | Offline | 94.39 | | 90.31 | | 68.72 | |
| | OCL | 59.63±3.44 | 35.37±4.47 | 57.12±4.46 | 35.41±6.74 | 55.92±6.37 | 38.17±3.75 |
| | EWC | 68.53±3.71 | 29.88±4.11 | 68.48±3.07 | 33.09±3.94 | **62.40±4.01** | 31.49±1.83 |
| | CLUDA | 45.02±1.36 | 32.71±3.73 | 40.53±0.89 | 33.59±1.56 | 38.41±0.91 | 34.54±1.70 |
| | AMBM | 75.99±2.54 | 29.40±3.49 | **72.41±2.29** | 29.05±3.06 | 57.39±2.19 | 27.05±2.03 |
| | CoPE | 40.13±0.9 | 27.07±2.38 | 39.55±1.53 | 26.4±2.07 | 38.06±1.56 | 26.3±1.96 |
| | OnPro | 28.71±2.29 | 29.51±2.61 | 28.62±1.07 | 29.22±2.83 | 27.72±1.87 | 28.05±3.82 |
| | OPWA | **86.1±1.87** | **47.07±4.45** | 71.45±5.75 | **47.51±4.33** | 54.38±1.26 | **42.01±3.21** |
| PPB-EMO | Offline | 95.12 | | 94.30 | | 90.05 | |
| | OCL | 80.03±3.13 | 44.84±8.91 | **81.56±2.13** | 39.58±15.98 | **79.82±2.39** | 42.41±7.49 |
| | EWC | **96.77±0.93** | 40.49±8.73 | 80.56±1.96 | 37.14±9.38 | 76.23±1.48 | 42.52±18.2 |
| | CLUDA | 46.95±0.10 | 48.44±3.28 | 42.94±0.21 | 43.92±3.92 | 45.67±2.99 | 50.4±5.99 |
| | AMBM | 57.89±1.14 | 33.14±1.76 | 55.58±1.86 | 31.46±7.08 | 57.13±2.59 | 31.38±3.21 |
| | CoPE | 63.91±4.11 | 33.7±5.1 | 63.19±1.97 | 35.38±9.97 | 62.16±1.8 | 33.39±4.72 |
| | OnPro | 37.11±4.68 | 39.7±5.19 | 38.42±2.56 | 39.89±5.51 | 37.28±1.8 | 44.14±8.04 |
| | OPWA | 86.65±0.99 | **62.95±8.89** | 77.94±3.18 | **49.97±6.94** | 67.4±2.0 | **53.67±7.92** |
| AMIGOS | Offline | 94.42 | | 88.14 | | 70.51 | |
| | OCL | 69.65±4.61 | 45.07±7.25 | 65.31±6.41 | 43.83±7.63 | 57.69±6.55 | 39.54±4.87 |
| | EWC | 63.42±4.14 | 30.24±2.77 | 63.21±2.59 | 34.46±4.07 | 56.86±2.50 | 29.66±3.15 |
| | CLUDA | 44.49±0.43 | 27.57±2.51 | 37.62±2.72 | 30.66±3.04 | 32.86±1.18 | 33.72±3.10 |
| | AMBM | 79.21±1.65 | 33.48±7.37 | 76.88±2.75 | 33.41±6.49 | 58.76±4.35 | 32.37±3.78 |
| | CoPE | 40.95±1.85 | 27.47±3.91 | 40.89±1.15 | 28.44±4.43 | 41.18±1.97 | 30.35±1.57 |
| | OnPro | 30.67±1.51 | 32.87±4.68 | 31.08±1.54 | 32.78±1.09 | 32.62±3.27 | 30.95±2.78 |
| | OPWA | **91.78±2.83** | **49.22±3.24** | **77.84±3.28** | **47.49±9.12** | **59.19±8.27** | **40.41±2.86** |

Table 1: Average Adaptation Accuracy (AAA) and Forgetting Mitigation Accuracy (FMA) (Mean ± Std) achieved on EEG datasets for different percentages of overlapping segments. Mean and standard deviation of five runs with varying subject sequences and initializations.

Table 1 provides a comparative analysis of different methods applied to EEG datasets and presents the AAA and FMA accuracies under three different overlap criteria for window segmentation. We evaluate the methods with 90%, 75% and 50% overlap. The results are derived from five experimental runs, each with different subject sequences and initializations. We report the mean and standard deviation of AAA and FMA across these runs. The proposed method, OPWA, consistently outperforms other methods in AAA across most of the datasets and settings, indicating its superior capability for rapid adaptation. On the BCI-IV-2a dataset, which consists of only 9 subjects with substantial amount of data samples per subject and reflects low inter-subject variability, EWC outperforms all other methods in both AAA and FMA. This is due to the small number of subjects with considerable amount of data, allowing EWC to effectively utilize the Fisher Information matrix for stable learning. OPWA, our proposed method, follows closely behind EWC, demonstrating strong adaptation capabilities and knowledge retention. However, as the datasets become more challenging with increased subject variability, such as in the DEAP, PPB-EMO, and AMIGOS datasets, EWC struggles to maintain high FMA performance, highlighting its limitations in real-time OCL scenarios. CLUDA performs reasonably well compared to other baselines, but still shows lower AAA and FMA scores. This underperformance may be due to its reliance on a pre-trained model on a source domain for adaptation, which is not ideal for the OCL setting where the model needs to continuously adapt without an explicit source domain. As a result, CLUDA struggles with the dynamic and evolving nature of the data, leading to reduced effectiveness in handling subject shifts in EEG data. OCL and AMBM show competitive performance in fast adaptation and even outperform OPWA on

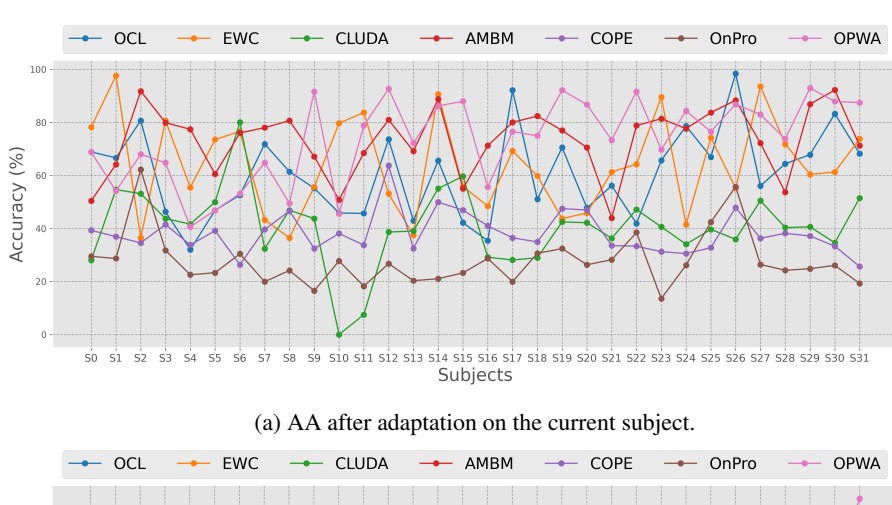

(a) AA after adaptation on the current subject.

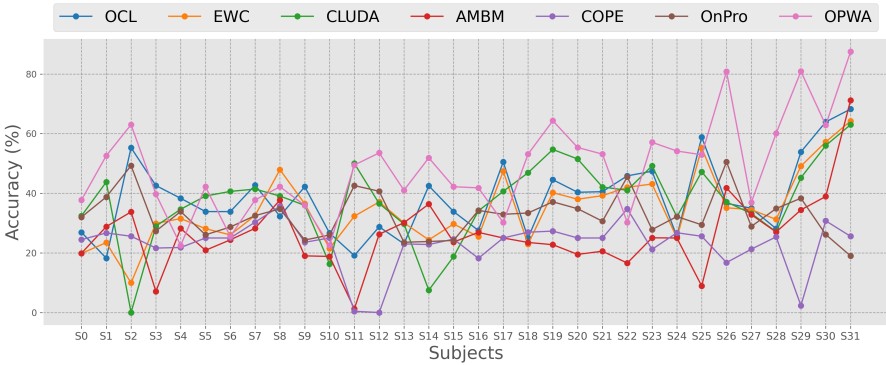

(b) FMA on previously seen subjects after learning on the current subject.

Figure 3: Adaptation Accuracy (AA) and Forgetting Mitigation Accuracy (FMA) as functions of training steps on online streams from the DEAP dataset.

the DEAP and PPB-EMO datasets. However, they do not maintain comparable levels of FMA. This indicates that while OCL performs remarkably well in fast adaptation, it struggles with retaining prior knowledge due to the lack of a dedicated forgetting mitigation strategy and therefore focuses more on robustness of adaptation. In contrast, AMBM incorporates a forgetting mitigation strategy in the meta-loop, but the multiple fast adaptation steps in the inner loop gradually reduce its overall impact. Approaches such as CoPE and OnPro show inconsistent performance, achieving reasonable AAA scores but lower FMA. This indicates possible limitations in their knowledge retention strategies. This inconsistency may stem from the fact that these methods are mainly tailored to image classification and class-incremental settings and do not fully address the challenges associated with subject shifts in EEG data. Moreover, it is noteworthy that the methods in the BCI-IV-2a have a higher FMA accuracy compared to AAA. This can be due to the fact that this dataset contains only 9 subjects, resulting in less inter-subject variability. Consequently, the new incoming data are closely aligned with the previously seen data distributions, which improves generalization. Now we turn our attention to Figure 3, which shows the subject-wise Adaptation Accuracy (AA) and the FMA for all previously seen subjects in the DEAP dataset. Figure 3a shows the adaptation performance of the alternatives on DEAP dataset. Note that the subjects arrive sequentially from left to right. Initially, EWC, OCL and AMBM lead in terms of AA, with their curves peaking at the beginning. However, as new subjects are introduced sequentially, OPWA begins to outperform these methods, particularly in later subjects. This suggests that OPWA performs better over time and demonstrates a stronger ability to adapt to new subjects as learning progresses. On the other hand, Figure 3b evaluates how well the model can maintain its performance on past subjects after it has adapted to the current subject. OPWA consistently outperforms the other methods in terms of FMA and shows superior generalization capabilities. It is able to effectively retain knowledge from most previous subjects, so its performance on past tasks remains strong even when new subjects are introduced. This shows that OPWA is able to retain knowledge and maintain high performance across multiple

| Method | 90% | | 75% | | 50% | |
|--------|-----|-----|-----|-----|-----|-----|
| | AAA | FMA | AAA | FMA | AAA | FMA |
| Offline | 96.25 | | 86.92 | | 63.01 | |
| OCL | 78.71 ± 10.55 | 46.19 ± 7.84 | 74.96 ± 7.70 | 44.99 ± 7.35 | **61.70 ± 4.36** | 41.65 ± 5.83 |
| EWC | 71.88 ± 9.64 | 33.39 ± 5.04 | 70.42 ± 6.63 | 31.18 ± 6.85 | 61.62 ± 2.94 | 33.82 ± 7.49 |
| CLUDA | 45.01 ± 3.08 | 27.57 ± 2.51 | 44.04 ± 1.10 | 28.05 ± 2.51 | 36.91 ± 2.34 | 31.42 ± 7.49 |
| AMBM | 57.41 ± 3.75 | 30.18 ± 2.04 | 57.74 ± 8.88 | 28.11 ± 2.42 | 50.64 ± 4.44 | 30.81 ± 1.23 |
| CoPE | 41.62 ± 1.39 | 28.14 ± 1.79 | 41.89 ± 3.01 | 28.68 ± 3.60 | 39.45 ± 2.47 | 29.99 ± 2.43 |
| OnPro | 30.94 ± 2.04 | 34.52 ± 2.73 | 29.93 ± 1.52 | 33.14 ± 2.27 | 31.48 ± 2.29 | 33.64 ± 2.98 |
| OPWA | **93.87 ± 2.39** | **54.90 ± 5.72** | **80.07 ± 4.49** | **50.60 ± 4.55** | 59.15 ± 3.34 | **43.61 ± 3.35** |

Table 2: AAA and FMA achieved on the AMIGOS (ECG) dataset at overlapping rates. Mean and standard deviation of five runs with varying subject sequences and initializations.

subjects, making it more robust during OCL. To further demonstrate the robustness of OPWA across different modalities, we conducted experiments on the AMIGOS (ECG) dataset and report the mean and standard deviation of 5 different runs in Table 2. OPWA shows superior performance on this dataset as well, indicating its ability to maintain consistent performance across different modalities.

**Discussion:** The OPWA method introduces a prototype memory buffer alongside the memory replay buffer to store the mean prototypes for each subject. Although this incurs additional storage costs, the memory requirement for prototypes is significantly lower than that of the replay buffer, as prototypes are lower-dimensional representations compared to the original data samples. Therefore, the total storage requirement remains manageable. The additional storage costs are justified by the advantage of preserving and aggregating compact representations of past subjects, which allows for better generalization without significantly increasing storage requirements.

The OPWA method computes pairwise distances between intra-class prototypes during the weighted aggregation process, which makes it more computationally intensive than baseline methods such as OCL, CoPE and OnPro. This pairwise distance calculation introduces quadratic complexity as it involves iterating over all pairs of prototypes within the same class. As a result, the computational cost of OPWA depends on both the number of classes and subjects in the dataset. However, for datasets with relatively few subjects and classes, such as the PPB-EMO dataset (up to 40 subjects and 4 classes), this complexity remains manageable. To reduce computational complexity and memory requirements in OPWA, one possible approach is to remove similar prototypes from memory. If two prototypes have similar weights, one of them can be discarded by analyzing the weight matrix. This reduces memory requirements and improves generalization by eliminating redundant prototypes. OPWA is more computationally efficient as compared to methods like EWC, CLUDA and AMBM. EWC requires computation of Fisher Information Matrix (FIM), which contains second-order information and is computationally intensive. Even with approximations thorough first order, the complexity remains high due to the large number of model parameters. CLUDA increases the complexity with multiple loss functions, including source and target domain losses and a discriminator loss, which increases the computational demands. AMBM, with its bilevel optimization process involving multiple inner- loop adaptation steps and a meta-loop for generalization, also incurs significant computational costs due to repeated gradient updates in both loops.

## 5 CONCLUSION

Our proposed method,OPWA, addresses the challenges of catastrophic forgetting in the presence of inter-subject variability in OCL, especially for physiological data. By leveraging the principles of prototypical networks, OPWA effectively retains the knowledge of previous subjects while adapting to new data streams. Our approach incorporates a robust prototype aggregation mechanism based on intra-class distance considerations, which ensures that the generalized prototypes accurately reflect the entire data distribution. This innovation goes beyond traditional momentum-based methods, providing a more accurate and reliable representation of class prototypes, especially in environments with significant subject shifts. The experimental results demonstrate the superior performance of OPWA in both AAA accuracy and FMA across different datasets, ensuring that the method is scalable to different modalities and different types of data sets.

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

# A APPENDIX

## A.1 DATASETS

**BCI IV-2a:** The BCI IV-2a dataset comprises data from 9 subjects, each undergoing 576 trials. Each trial consists of 22 channels over a temporal span of 400. Each trial is captured at a sampling rate of 250 Hz. In our experiments, the segment window size is 400 with three different ratios of overlap

which include 90%, 75% and 50%. The data are categorized into four different classes based on the types of motor imagery movements: left hand, right hand, tongue and both feet.

**DEAP:** The DEAP dataset consists of data from 32 subjects, each participated in 40 trials. The data are categorized into four classes based on the quadrants of valence and arousal: High arousal and high valence (HAHV), low arousal and low valence (LALV), high arousal and low valence (HALV), and low arousal and low valence (LALV). To improve data quality, we exclude first 13 seconds of each trial and divided the trials into 32 channels over a temporal span of 768. The segments overlap with a step size of 128, 192, 384, resulting in 45, 30, and 15 segments per trial, respectively.

**AMIGOS:** The AMIGOS dataset includes EEG, ECG and GSR recordings from wearable sensors in two scenarios: 40 participants watched 16 short emotional videos alone and later four longer videos either alone or in groups. We utilized a pre-processed version of the dataset, excluding subjects with missing data, specifically subjects $\{4, 5, 8, 10, 11, 22, 24, 25, 26, 28, 30, 31, 32, 40\}$. Data were sampled at 128 Hz, and 5-second segments were created for each trial at 90%, 75% and 50% of overlapping. Participants rated their emotional responses on a continuous scale from 1 (low) to 9 (high) for arousal, valence and dominance. Each subject's self-assessment scores were then used to classify valence and arousal.

**PPB-Emo:** The PPB-Emo dataset comprises data from 40 participants engaged in driving tasks while experiencing various emotions. During the experiment, drivers watched emotion-evoking videos and then performed driving tasks reflecting those emotions. The dataset captures multiple modalities for emotion recognition, including behavioral data, facial videos, body gesture data, and physiological signals. Specifically, it includes 32-channel EEG data recorded at 250 Hz using the EnobioNE, a wireless EEG device. Additionally, participants provided self-reported ratings of valence, arousal, and dominance for each emotional state on a 9-point scale (1 = "not at all" to 9 = "extremely").

## A.2 EXPERIMENTAL SETUP

In our experiments, we utilize the Adam optimizer with a learning rate of 0.0001, along with a learning rate scheduler that employs a step size of 30 and a decay rate of 0.9. The batch size is set to 32 samples, and an equal number of samples are fetched from the memory buffer using a reservoir sampling technique. For prototype evolution in CoPE and within-subject training in OPWA, we use a momentum of 0.9. Additionally, the parameter $\sigma$ in the Gaussian kernel is set to 1. Affine transformations are applied for data augmentation in CoPE and OnPro for contrastive losses.

After each subject shift, OPWA stores the prototypes of the current subject in the prototype memory and performs an aggregation over the prototypes of the previously seen subjects. We work with a subject-aware setting where the subject shifts are known. The subject-agnostic setting, where the shifts are unknown, is reserved for future work. However, our method can be easily integrated with existing subject-agnostic approaches Duan et al. (2024b) that use a detection mechanism to identify subject shifts during online learning.

## A.3 ADDITIONAL EXPERIMENTS

Figure 4 illustrates the comparison between the embedding spaces learned by the AMBM and OPWA approaches. Both models were trained in an OCL setting using the AMIGOS EEG dataset. At the end of the training phase, the embedding vectors of the test data were extracted from the trained models and visualized using UMAP dimensionality reduction. In the embedding space learned by AMBM, the vectors of the different classes are densely packed, leading to significant overlap between them. This lack of separation poses a challenge for the classifier, as the absence of clear class boundaries makes accurate classification difficult. In contrast, the embedding space generated by OPWA shows clear and distinct class boundaries, with samples from different classes well separated from each other. This improved separation facilitates better discrimination between classes and highlights OPWA's superior performance in learning discriminative and separable embeddings space.

## A.4 ALGORITHM

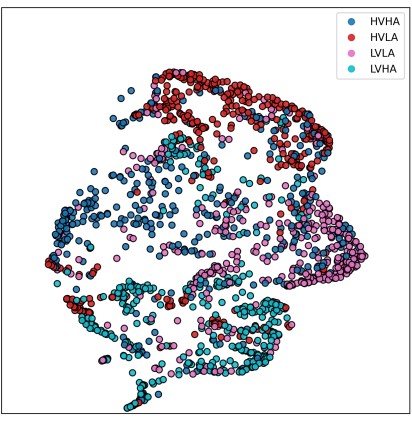 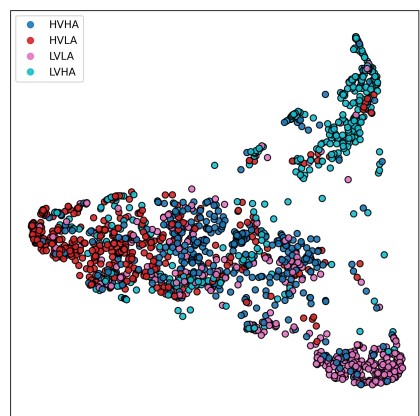

(a) Embedding space learnt with AMBM        (b) Embedding space learnt with OPWA

Figure 4: UMAP visualization of embedding space learnt using AMBM and the proposed OPWA approach.

---

**Algorithm 1** OPWA

1: **Input:** Sequence of subjects $\mathcal{S} = \{\mathcal{S}_1, \mathcal{S}_2, \ldots, \mathcal{S}_n\}$, Memory buffer $\mathcal{M}$, Model parameters $\theta_f, \theta_g, \theta_\phi$
2: **Initialize:** Prototype memory buffer $\mathcal{M}_p$
3: **for** each subject $\mathcal{S}_i \in \mathcal{S}$ **do**
4:     **for** each mini-batch data $\mathcal{D}^t_{\mathcal{S}_i}$ from subject $\mathcal{S}_i$ **do**
5:         **Retrieve replay buffer data:** $\mathcal{D}_b$ from $\mathcal{M}$
6:         **Combine current and buffer data:** $\mathcal{D} = \mathcal{D}^t_{\mathcal{S}_i} \cup \mathcal{D}_b$
7:         **Compute embedding vectors:** $\boldsymbol{z} = g(f(\mathcal{D}; \theta_f); \theta_g)$
8:         **Prototype Calculation:** Compute class prototypes using equation Equation 2
9:         Perform training using Equation 10
10:     **end for**
11:     **Prototype Update:** Update proto after minii-batch using Equation 4
12: **end for**
13: **Subject Shift:** When subject $\mathcal{S}_i$ shifts to $\mathcal{S}_{i+1}$
14: **for** each class $k$ **do**
15:     Compute weights using equation 6
16:     **Weight Normalization:** Normalize the weights using Equation 7
17:     **Prototype Aggregation:** Aggregate prototypes stored in $\mathcal{M}_p$ using Equations8
18: **end for**
19: **Update Prototypes:** Store aggregated prototypes $\hat{\mathcal{P}}_k$ and proceed with the next subject
20: **Output:** Trained model

---

