# OpenReview forum: "Online Sequential Learning from Physiological Data with Weighted Prototypes: Tackling Cross-Subject Variability"
_ICLR.cc/2025/Conference — Submitted to ICLR 2025_

### Official Review · Reviewer_Jz9u · 2024-10-27

**Soundness:** 3
**Presentation:** 3
**Contribution:** 2
**Rating:** 5
**Confidence:** 4

**Summary:**

This paper proposes a method called Online Prototype Weighted Aggregation (OPWA) for online sequential learning, aimed at addressing the issue of cross-subject variability in online continual learning. By introducing a novel prototype-weighted aggregation mechanism, the paper effectively mitigates this challenge.

**Strengths:**

1.Appropriate solution to the problem: The paper combines cross-subject prototype weighting and momentum-based prototype updating, accounting for dynamic adaptation to distribution shifts and robustness to outliers.
2.Comprehensive validation: The OPWA method was validated on four datasets, demonstrating its applicability across both EEG and ECG data.

**Weaknesses:**

1.Limited methodological innovation: Cross-subject prototype weighting and momentum-based prototype updating are common techniques. The contribution of this paper lies more in the combination of these methods rather than in any novel algorithmic development.
2.Need for clarification of online learning concepts: Concepts like incremental learning and continual learning are closely related to online learning. It is recommended that the authors provide a more detailed comparison and explanation of these terms in the paper.
3.High model complexity and computational cost: The prototype-weighted aggregation method introduced in the paper incurs additional computational overhead, particularly with the complex distance calculations and weight normalization involved in cross-subject prototype aggregation. It would be helpful if the authors could compare the computational cost of their method with others.
4.Limited experimental coverage: Although the paper includes experiments on several public physiological datasets, the sample size and diversity are limited, especially in terms of the number of subjects, which may not fully validate the generalizability of the method. Additionally, the DEAP and AMIGOS datasets used in the paper rely on subjective self-reported emotional evaluations, which may introduce significant labeling noise.
5.Lack of reproducibility: The code used in the paper has not been made publicly available, hindering reproducibility.

**Questions:**

Please see Weaknesses.

---

> ### Author Response · Authors · 2024-11-28
> **Response to Reviewer Jz9u**
>
> **Q1:** Limited methodological innovation: Cross-subject prototype weighting and momentum-based prototype updating are common techniques. The contribution of this paper lies more in the combination of these methods rather than in any novel algorithmic development.
>
> **R1:** Thank you for your feedback. We understand your concerns about the novelty of our approach and would like to highlight the key innovations of our work in response to your comment.
>
> The core strength of our method lies in the integration of prototype aggregation with normalized weights, ensuring that subject-specific prototypes contribute appropriately to the overall class representation without being biased by any single prototype. A key aspect of the novelty of our approach is the formulation of intra-class prototype weighting, which extends beyond traditional momentum-based updates. While momentum-based methods, such as CoPE, gradually update prototypes to accommodate new data streams, they often drift away from representations of past subjects over time, increasing the risk of forgetting. In contrast, our approach incorporates a weighted aggregation scheme that adapts to distributional shifts between subjects and ensures a more robust representation of class prototypes. This dynamic weighting, determined by the distances between prototypes, allows us to prioritize those that are more stable and reflect the true data distribution. To the best of our knowledge, this weighted aggregation strategy has not been previously employed in the context of OCL. Our proposed method provides a refined approach for managing prototypes, especially in scenarios with significant subject shifts, and represents a substantial advancement over existing techniques.
>
> We have modified the introduction section to highlight the shortcomings of existing prototype based OCL methods. Please refer to the revised version, Section 1 (Page 2 Lines 091-103).
>
> **Q2:** Need for clarification of online learning concepts: Concepts like incremental learning and continual learning are closely related to online learning. It is recommended that the authors provide a more detailed comparison and explanation of these terms in the paper.
>
> **R2:** Thank you for your valuable feedback. We have briefly described the concepts of incremental learning, continual learning, and online learning in the Introduction section of the revised version for better clarity. Please refer to Page 1, Line 050.
>
> **Q3:** High model complexity and computational cost: The prototype-weighted aggregation method introduced in the paper incurs additional computational overhead, particularly with the complex distance calculations and weight normalization involved in cross-subject prototype aggregation. It would be helpful if the authors could compare the computational cost of their method with others.
>
> **R3:** Thank you for raising the concern on the computational complexity of the proposed method. In the revised version of the manuscript, we have included a detailed discussion of the computational demands of the proposed framework. We have compared the complexity of OPWA with existing methods and provided insights into scaling with the number of subjects and classes. Furthermore, we discuss possible strategies to reduce the computational load, such as removing similar prototypes and optimizing memory usage.
>
> Please refer to revised version Section 4.1, Page 10, Lines 502-525.

---

> > ### Author Response · Authors · 2024-11-28
> > **Response to Reviewer Jz9u**
> >
> > **Q4:** Limited experimental coverage: Although the paper includes experiments on several public physiological datasets, the sample size and diversity are limited, especially in terms of the number of subjects, which may not fully validate the generalizability of the method. Additionally, the DEAP and AMIGOS datasets used in the paper rely on subjective self-reported emotional evaluations, which may introduce significant labeling noise.
> >
> > **R4:** Thank you for your valuable feedback. In the revised version of the manuscript, we have strengthened the experimental evaluation by conducting five independent experiments with random initialization and varying subject sequences to account for possible variability in the results, and we have now reported the mean and standard deviation of the performance metrics in Section 4.1 Table 1 and 2, and Line 417.
> >
> > Acquiring large and diverse datasets poses significant challenges due to privacy concerns, organizational and legal constraints, and the need for substantial human resources for data annotation. Despite these limitations, the proposed method exhibited consistent performance across two distinct datasets (DEAP and AMIGOS), underscoring its robustness. Furthermore, the use of cross-validation and experimentation on multiple datasets helped address concerns about overfitting to specific datasets or limited sample diversity. Similar challenges related to dataset size and labelling have been documented in related studies [1,2] emphasizing the broader necessity for more comprehensive datasets in this field.
> >
> > [1] Pais, D., Brás, S., & Sebastião, R. (2024). Overcoming the Small Dataset Challenge in Healthcare. 2024 IEEE 22nd Mediterranean Electrotechnical Conference (MELECON), 497-502.
> >
> > [2]Jiang, W., Lan, Y., & Lu, B. (2024). REmoNet: Reducing Emotional Label Noise via Multi-regularized Self-supervision. ACM Multimedia.
> >
> >
> >
> >
> > **Q5:** Lack of reproducibility: The code used in the paper has not been made publicly available, hindering reproducibility.
> >
> > **R5:** Thank you for your comment regarding reproducibility. To preserve anonymity during the review process, we have not made the code publicly available at this stage. However, we have attached the code as supplementary material, and we intend to release it on a public GitHub repository upon acceptance of the paper.

---

### Official Review · Reviewer_qM24 · 2024-10-30

**Soundness:** 2
**Presentation:** 2
**Contribution:** 2
**Rating:** 3
**Confidence:** 3

**Summary:**

This paper presents a novel method, OPWA, to address the problem of catastrophic forgetting in the presence of inter-subject variability through the use of prototypical networks. The method considers the underlying data distribution of each subject while constructing global prototypes. The aggregation mechanism takes into account intra-class variance and thereby naturally improves the decision boundaries between inter-class prototypes. The experiment results show that the proposed method achieve superior performance in both adaptation and forgetting mitigation across different modalities.

**Strengths:**

The proposed method can address the problem of catastrophic forgetting in the presence of inter-subject variability through the use of prototypical networks.

The experiment results highlight the superior performance in both adaptation and forgetting mitigation across different modalities.

**Weaknesses:**

The technical innovation of the paper is questionable. It appears that the paper does not introduce a novel methodology but rather incorporates the prototypical approach into Online Continual Learning, leveraging the derivation of generalized prototypes that represent the entire data distribution to ensure generalizability in the face of cross-subject variability. Although this approach has a slight novelty, it is not substantial.

The experiments are not solid.  As a continual learning study, the paper only conducts a single experiment without multiple repetitions, and does not swap the order of subjects across multiple experiments, making it difficult to exclude the influence of randomness on the results. Statistical tests are lacking to ensure whether the proposed method demonstrates a significant performance improvement over existing methods.

There is a lack of discussion on the computational demands of the framework, which could be significant given the continual adaptation process and may limit its deployment in resource-constrained settings, particularly in a cross-subjects online continual learning setup.

The presentation of the paper still has considerable room for improvement. For instance: There is a formatting error in the title, the introduction to related work is overly lengthy, the sizing of the figures and tables is inappropriate (Table 2 is excessively large, while Figure 3 is too small), and there are issues with the formatting of the titles.

**Questions:**

Please see Weakness.

---

> ### Author Response · Authors · 2024-11-28
> **Response to Reviewer qM24**
>
> **Q1:** The technical innovation of the paper is questionable. It appears that the paper does not introduce a novel methodology but rather incorporates the prototypical approach into Online Continual Learning, leveraging the derivation of generalized prototypes that represent the entire data distribution to ensure generalizability in the face of cross-subject variability. Although this approach has a slight novelty, it is not substantial.
>
> **R1:** Thank you for your feedback. We understand your concerns about the novelty of our approach and would like to highlight the key innovations of our work in response to your comment.
>
> The core strength of our method lies in the integration of prototype aggregation with normalized weights, ensuring that subject-specific prototypes contribute appropriately to the overall class representation without being biased by any single prototype. A key aspect of the novelty of our approach is the formulation of intra-class prototype weighting, which extends beyond traditional momentum-based updates. While momentum-based methods, such as CoPE, gradually update prototypes to accommodate new data streams, they often drift away from representations of past subjects over time, increasing the risk of forgetting. In contrast, our approach incorporates a weighted aggregation scheme that adapts to distributional shifts between subjects and ensures a more robust representation of class prototypes. This dynamic weighting, determined by the distances between prototypes, allows us to prioritize those that are more stable and reflect the true data distribution. To the best of our knowledge, this weighted aggregation strategy has not been previously employed in the context of OCL. Our proposed method provides a refined approach for managing prototypes, especially in scenarios with significant subject shifts, and represents a substantial advancement over existing techniques.
>
> We have modified the introduction section to highlight the shortcomings of existing prototype based OCL methods. Please refer to the revised version, Section 1 (Page 2 Lines 091-103).
>
> **Q2:** The experiments are not solid. As a continual learning study, the paper only conducts a single experiment without multiple repetitions, and does not swap the order of subjects across multiple experiments, making it difficult to exclude the influence of randomness on the results. Statistical tests are lacking to ensure whether the proposed method demonstrates a significant performance improvement over existing methods.
>
> **R2:** Thank you for your valuable feedback. In the revised version of the manuscript, we have strengthened the experimental evaluation by conducting five independent experiments with random initialization and varying subject sequences to account for possible variability in the results, and we have now reported the mean and standard deviation of the performance metrics in Section 4.1 Tables 1 and 2. Please refer to revised version Page 8 and 10, and Line 416.
>
> **Q3:** There is a lack of discussion on the computational demands of the framework, which could be significant given the continual adaptation process and may limit its deployment in resource-constrained settings, particularly in a cross-subjects online continual learning setup.
>
> **R3:** Thank you for highlighting this concern. In the revised version of the manuscript, we have included a detailed discussion of the computational demands of the proposed framework in Section 4.1 Page 10 Line 502. We have compared the complexity of OPWA with existing methods and provided insights into scaling with the number of subjects and classes. Furthermore, we discuss possible strategies to reduce the computational load, such as removing similar prototypes and optimizing memory usage.
>
> **Q4:** The presentation of the paper still has considerable room for improvement. For instance: There is a formatting error in the title, the introduction to related work is overly lengthy, the sizing of the figures and tables is inappropriate (Table 2 is excessively large, while Figure 3 is too small), and there are issues with the formatting of the titles.
>
> **R4:** Thank you for your feedback. We have improved the presentation of the paper based on your suggestions. We have removed the excessive information from the related work section and adjusted the presentation of the tables and figures.

---

> > ### Comment · Reviewer_qM24 · 2024-11-28
> >
> > Thank you for the rebuttal. The paper has demonstrated some improvement; however, I have a remaining question regarding the motivation for using Online Continual Learning (OCL) in EEG tasks. The authors state in the paper:
> >
> > *"For instance, publicly available datasets are usually collected in specific environments using specialized emotion elicitation techniques, which often cannot be generalized to all situations (Khare et al., 2024). In addition, these datasets usually represent a limited number of classes (Duan et al., 2024b). Dynamic and time-critical environments lead to different levels of stress in individuals, which can vary considerably from one environment to another. This limitation makes it necessary to train models online on continuous data streams, a process known as Online Continual Learning (OCL)."*
> >
> > However, the authors have applied OCL only to address **subject shift**, and all experiments are conducted within **a single dataset without involving cross-paradigm EEG data collection.** This seems not to fully align with the dynamic EEG acquisition scenarios described in the motivation. Could the authors elaborate on this?

---

> > > ### Author Response · Authors · 2024-12-02
> > > **Response to Reviewer qM24**
> > >
> > > We appreciate your question about the rationale for using Online Continual Learning (OCL) in EEG tasks and its alignment with dynamic EEG acquisition scenarios.
> > >
> > > We acknowledge that this work does not address cross-paradigm EEG data acquisition, which is indeed a critical and challenging research direction. However, the primary focus of this work is to tackle the problem of inter-subject variability (Revised version Lines: 074 - 079) within a single EEG dataset, a challenge that is widely recognized in the field [1,2,3]. Studies such as the [1,2,3] have demonstrated the importance of addressing this problem in similar contexts to improve the generalization of models across different subjects within a consistent data collection paradigm.
> > >
> > > Although cross-paradigm EEG acquisition is outside the scope of this work, we acknowledge its importance and recognise it as an important direction for future research. The motivation presented in this work reflects the broader applicability of OCL in dynamic environments (Revised version Lines 047 - 050), and we see our current work as an important step towards this vision by addressing inter-subject variability (Revised version: Lines 074 - 079).
> > >
> > >
> > > [1] Retain and adapt: Online sequential eeg classification with subject shift. IEEE Transactions on Artificial Intelligence, 5:4479–4492, 2024
> > >
> > > [2] Online continual decoding of streaming eeg signal with a balanced and informative memory buffer. Neural networks : the official journal of the International Neural Network Society, 176:106338, 2024.
> > >
> > > [3] Replay with Stochastic Neural Transformation for Online Continual EEG Classification, BIBM 2023

---

> > > > ### Comment · Reviewer_qM24 · 2024-12-03
> > > >
> > > > Thank you for the detailed clarification, which highlights the paper's focus on addressing inter-subject variability within a single EEG dataset and its alignment with OCL's broader applicability. Your explanation effectively distinguishes the scope of this work from future directions, such as cross-paradigm EEG acquisition.
> > > >
> > > > To further enhance clarity and avoid potential misinterpretation of the motivation, I suggest revising the **Introduction** to explicitly delineate the current focus and future research directions. This adjustment would strengthen the presentation of the study's contributions.
> > > >
> > > > From my perspective as a reviewer, while this work demonstrates certain contributions, the methodological innovation and analysis of why the approach works may not fully meet ICLR's standards. I strongly encourage the authors to further investigate the role of the proposed method in the OCL process, moving beyond basic feature visualizations. Additionally, a more comprehensive evaluation would be highly beneficial for strengthening the work.

---

### Official Review · Reviewer_WA8F · 2024-11-02

**Soundness:** 3
**Presentation:** 2
**Contribution:** 2
**Rating:** 5
**Confidence:** 3

**Summary:**

This paper presents a new method for online continual learning. The presented method, Online Prototype Weighted Aggregation (OPWA), is specifically designed for catastrophic forgetting in the presence of inter-subject variability through the use of prototypical networks.
The method employs a prototype aggregation mechanism that fuses intra-class prototypes into generalized representations by accounting for both within-class and inter-class variation between subjects. The authors conducted experiments on different modalities using 4 datasets.

**Strengths:**

The main focus of the paper, catastrophic forgetting, is important in machine learning. The subject variability in bio-signals is a big obstacle to the deployment of the ML algorithms in a wide range of bio-signal applications, making the authors' motivations timely and solid. The paper is well-written and easy to follow.

**Weaknesses:**

This paper focuses on the catastrophic forgetting problem from the inter-subject variability of bio-signals, especially EEG. Therefore, the problem the authors focus on solving is very similar to domain adaptation/generalization. Since domain adaptation /generalization aims to train the models in a way that when the domain (mostly subjects in bio-signals) changes, the model performs well. Although the authors here formulate the problem from the training (multiple subjects/domains during training), I feel like there should be some baseline comparisons with domain adaptation/generalization works [3-4].


The catastrophic forgetting is generally defined as the phenomenon of forgetting previously learned information when trained on new data. And, the model evaluation is performed when it is forced to learn multiple tasks sequentially. In this paper, sequential learning is formulated subject-wise instead of task-wise, which makes the evaluation and comparison controversial as the domain generalization/adaptation tools can also be used for subject-wise sequential learning.

[1] James Kirkpatrick, Overcoming catastrophic forgetting in neural networks, PNAS 2016.

[2] Sara Babkniya, A Data-Free Approach to Mitigate Catastrophic Forgetting in Federated Class Incremental Learning for Vision Tasks, NeurIPS 2023.

I believe that it would strengthen the paper to include a baseline comparison with existing domain adaptation and generalization methods, particularly those applied to bio-signals across different subjects.
Since the proposed approach has similarities with domain adaptation/generalization—aiming for robustness to inter-subject variability—evaluating the method against these baselines could contextualize its unique advantages and potential trade-offs.


[3] Theo Gnassounou, Convolution Monge Mapping Normalization for learning on sleep data. NeurIPS 2023

[4] Yilmazcan Ozyurt, Contrastive Learning for Unsupervised Domain Adaptation of Time Series. ICLR 2023

**Questions:**

1) Is there any ablation studies performed by the authors regarding the sequence of subjects, similar to the sequence of tasks?

2) Why not focus on domain adaptation or generalization but sequential subject learning? Are there any preliminary experiments showing this approach is superior? The authors can include a brief discussion in the paper comparing their sequential subject learning approach to domain adaptation, highlighting the key differences and potential advantages.

---

> ### Author Response · Authors · 2024-11-28
> **Response to Reviewer WA8F**
>
> **Q1:** This paper focuses on the catastrophic forgetting problem from the inter-subject variability of bio-signals, especially EEG. Therefore, the problem the authors focus on solving is very similar to domain adaptation/generalization. Since domain adaptation /generalization aims to train the models in a way that when the domain (mostly subjects in bio-signals) changes, the model performs well. Although the authors here formulate the problem from the training (multiple subjects/domains during training), I feel like there should be some baseline comparisons with domain adaptation/generalization works [3-4].
>
> **R1:** Thank you for the suggestion. In the revised version, we included two domain adaptation/generalization baselines, specifically [1] and [4], as these align more closely with our methodology. Please refer to revised version Page 7 Lines 347-356. We chose not to include [2], as it focuses on privacy concerns in federated learning and employs generative models to produce synthetic data, which is not relevant to our setup. Our approach leverages a memory buffer of real data samples, avoiding the additional computational cost of training a generative model. Additionally, we found [3] to have high computational demands, making it less suitable for our real-time OCL constraints.
>
> We include [4] by treating the memory buffer as source data. In our experiments, we pretrained the model using memory buffer samples before sequentially training it on each subject. However, this approach was limited by the lack of a rich, diverse source dataset in the OCL setting, which resulted in significant catastrophic forgetting.
>
> Please refer to Page 3 Lines 153-161 for the limitations of domain adaptation baselines. Performance of domain adaptation baselines are discussed in section 4.1 Page 8 Lines 420- 430.
>
> **Q2:** The catastrophic forgetting is generally defined as the phenomenon of forgetting previously learned information when trained on new data. And, the model evaluation is performed when it is forced to learn multiple tasks sequentially. In this paper, sequential learning is formulated subject-wise instead of task-wise, which makes the evaluation and comparison controversial as the domain generalization/adaptation tools can also be used for subject-wise sequential learning.
>
> **R2:** Thank you for highlighting the potential resemblance between our problem and domain adaptation. However, we emphasize significant differences between OCL with sequentially arriving subjects and traditional domain adaptation approaches.
>
> A key distinction is the lack of rich source data for pretraining in OCL. In domain adaptation, models are typically pretrained on a large and diverse source dataset, allowing them to capture robust representations that are later fine-tuned on downstream tasks. In contrast, OCL operates under memory constraints, where storing all previous data is infeasible. The model incrementally learns representations from small, sequentially arriving data chunks, making it more susceptible to overfitting due to the absence of a comprehensive source dataset.
>
> Although domain generalization techniques can be applied in this setting, they are often designed to fine-tune pretrained models on downstream tasks and may not directly adaptable in OCL. To validate this, we included two domain adaptation baselines as suggested. In these experiments, the current memory data was treated as source data for pretraining after each subject shift, and the model was fine-tuned on the current data stream. However, results indicated significant catastrophic forgetting, as the model struggled to retain generalized knowledge from the limited and dynamic source data after training on new tasks. This highlights the unique challenges of OCL that domain adaptation approaches are not fully equipped to address. We further highlighted the key differences between OCL and domain adaptation in related works. Please refer to revised version Page 3 Lines 153-161.

---

> > ### Author Response · Authors · 2024-11-28
> > **Response to Reviewer WA8F**
> >
> > **Q3:** I believe that it would strengthen the paper to include a baseline comparison with existing domain adaptation and generalization methods, particularly those applied to bio-signals across different subjects. Since the proposed approach has similarities with domain adaptation / generalization, aiming for robustness to intersubject variability, evaluating the method against these baselines could contextualize its unique advantages and potential trade-offs.
> >
> > **R3:** Thank you for the suggestion. In the revised version, we have included two domain adaptation/generalization baselines, [1] and [4], to address inter-subject variability and evaluate the performance of the proposed approach in comparison. Please refer to revised version Page 7 Lines 347-356.  The section 4.1 (Page 8 Lines 420-430) now provides a detailed analysis of the performance of these methods, along with a discussion of their limitations in the context of OCL. Please refer to revised version Page 10 Lines 518-525 for detailed discussion on computational complexity.
> >
> >
> > **Q4:** Is there any ablation studies performed by the authors regarding the sequence of subjects, similar to the sequence of tasks?
> >
> > **R4:** Thank you for your question. In the previous version, subjects were arranged sequentially based on their IDs. In response to the reviewer’s concern regarding the sequence of subjects, we performed five experiments with different random seeds, each creating a varied random arrangement of subjects. We then report the mean and standard deviation of the evaluation metrics across these experiments in Tables 1 and 2 (Page 8 and 9 , Line 416). This approach helps account for the influence of different subject sequences on the performance of the methods.
> >
> > **Q5:** Why not focus on domain adaptation or generalization but sequential subject learning? Are there any preliminary experiments showing this approach is superior? The authors can include a brief discussion in the paper comparing their sequential subject learning approach to domain adaptation, highlighting the key differences and potential advantages.
> >
> > **R5:** Thank you for the question. In this work, we specifically focus on OCL as it is directly applicable to real-world scenarios involving bio-signals, such as EEG, where new subjects are continuously added to the system. As highlighted in the introduction, this approach is tailored to environments where data is introduced incrementally and in real-time, and the model must adapt to each new subject while retaining the knowledge of the previous subjects.
> >
> > OCL differs from traditional domain adaptation and generalization because it does not rely on a large, pre-existing source dataset for pre-training. OCL involves learning from new subjects without access to a rich, static dataset, and the model must learn incrementally with limited memory. This makes our approach more suitable for scenarios where data is scarce, such as EEG classification in real-time applications.
> >
> > In the revised manuscript, we discuss these differences and emphasize the unique advantages of OCL approach, especially in situations with limited data and memory constraints. Please refer to Section 4.1 Discussion, Page 10, Lines 519- 527. Limitation of domain adaptation techniques are also highlighted in section 3 Page 3 Lines 153-161.

---

### Official Review · Reviewer_qNPd · 2024-11-05

**Soundness:** 3
**Presentation:** 3
**Contribution:** 3
**Rating:** 6
**Confidence:** 5

**Summary:**

The work tackles the problem of online sequential learning from physiological data by introducing a a robust prototype aggregation mechanism that synthesizes global prototypes from cross-subject prototypes. It considers the underlying data distribution of each subject while constructing global prototypes. The aggregation mechanism takes into account intra-class variance and thereby naturally improves the decision boundaries between inter-class prototypes.

**Strengths:**

The work is tackling an important problem and the paper is easy to follow in general.

The proposed method fits the problem well by considering the subject variance and distribution shift in generating the prototypes.

The authors performed detailed experiments on top of numerous public EEG datasets and demonstrated the effectiveness of proposed method.

**Weaknesses:**

1. Given the work tackles specifically the EEG classification task, the authors need to highlight in introduction of the possible application scenarios for the proposed online EEG classification algorithm in real world.
2. It is unclear how the subjects are arranged sequentially, is it arranged by subject id or other attributes? would the author consider difference sequential orders?
3. Would be better if the author could write 1-2 paragraphs in related work to summarize the major strengths and weaknesses of current online decoding approaches specifically in EEG analysis domain, in addition to the introduciton on general continual learning approaches.
4. At the beginning of Section 3, before going into the technical details of the methods/algorithms, please write a summary paragraph about the main feature/benefits/strengths of the proposed approach.
5. It is recommended that the author to provide more detailed explanations and explanations for Fig 3 and 4.
6. In the conclusions, in addition to summarizing the actions taken and results, please strengthen the explanation of their significance. The authors are suggested to enlist limitations of the study in the conclusion also.

Other minors:
line 174, as reinforce learning have very specific meaning, I would suggest change it to something like “enforce learning of past subjects”.
Some of the datasets are for motor imagery and some are for emotion recognition, better to mention the purpose of each dataset in 4.1 description.

**Questions:**

As listed in strength and weakness section.

---

> ### Author Response · Authors · 2024-11-28
> **Response to Reviewer qNPd**
>
> **Q1:** Given the work tackles specifically the EEG classification task, the authors need to highlight in introduction of the possible application scenarios for the proposed online EEG classification algorithm in real world.
>
> **R1:** In the revised version, we have highlighted real-world scenarios, including air traffic management, aviation, entertainment, and search and rescue operations, where physiological signals have been classified for tasks such as mental workload monitoring and decision-making, and can further benefit from the proposed OCL approach to adapt in such dynamic environments. Please refer to revised version Section 1 (Lines 053-073 ).
>
> **Q2:** It is unclear how the subjects are arranged sequentially, is it arranged by subject id or other attributes? would the author consider difference sequential orders?
>
> **R2** In the previous version, subjects were arranged sequentially based on their IDs. In the revised version, we perform experiments with five different runs, each using a random sequence of subjects, and report the mean and standard deviation of accuracy across these runs in Section 4.1 Table 1 and Table 2. Please refer to revised version Page 8 and 10, and Line 416.
>
> **Q3** Would be better if the author could write 1-2 paragraphs in related work to summarize the major strengths and weaknesses of current online decoding approaches specifically in EEG analysis domain, in addition to the introduciton on general continual learning approaches.
>
> **R3** Thank you for the suggestion. In the revised version, we have updated the Related work section to specifically address online EEG decoding approaches. We summarize the strengths and weaknesses of recent methods, including AMBM, which perform well in fast adaptation using contrastive loss but suffers from overfitting and high computational cost due to its bilevel optimization nature. We also discuss memory rehearsal approaches, such as those using clustering for balanced memory representation, and highlight their limitations when processing noisy or non-stationary data. In addition, we have included a discussion on domain adaptation based approaches and their limitations, especially in dynamic and data-scarce environments. Please refer to Section 2 Lines 135-161 of the revised version.
>
> **Q4:** At the beginning of Section 3, before going into the technical details of the methods/algorithms, please write a summary paragraph about the main feature/benefits/strengths of the proposed approach.
>
> **R4:** Thank you for the suggestion. We have revised section 3 to include a summary paragraph that highlights the key strengths of our approach and emphasizes its novelty by addressing the limitations of existing methods. Please refer to the revised version, Section 3 Lines 173-183.
>
> **Q5** It is recommended that the author to provide more detailed explanations and explanations for Fig 3 and 4.
>
> **R5:** In the revised version, we have included more detailed explanations for Figures 3 and 4. These additional explanations elaborate on the performance trends observed across different methods during training and how the results highlight the strengths of OPWA in terms of both adaptation (AA) and forgetting mitigation (FMA). Please refer to the Page 9, Lines 474-485 and Page 14 Line 743.
>
> **Q6:**  In the conclusions, in addition to summarizing the actions taken and results, please strengthen the explanation of their significance. The authors are suggested to enlist limitations of the study in the conclusion also.
>
> **R6:**  Thank you for your suggestion. In the revised version, we have strengthened the conclusion by highlighting the significance of our proposed method. We have also added a separate section in which we discuss the limitations of our study. Please refer to the revised version Page 10, Lines 502-525 and Lines 534-538.
>
> **Q7:** Other minors: line 174, as reinforce learning have very specific meaning, I would suggest change it to something like “enforce learning of past subjects”. Some of the datasets are for motor imagery and some are for emotion recognition, better to mention the purpose of each dataset in 4.1 description.
>
> **R7:** Thank you for your suggestions. In the revised version of the manuscript, we have updated the term "reinforce learning" to "enforce learning of past subjects" Page 4 Line 191. Additionally, we have clarified the purpose of each dataset in Section 4.1. Page 7 Lines 329-331.

---

### Author Response · Authors · 2024-11-28
**General Comments**

We thank the reviewers for their insightful comments and suggestions, which have helped us improve the quality of our manuscript. Below, we provide detailed responses to each comment. All changes have been incorporated in the revised manuscript, and these are highlighted in red for your convenience.

We have carefully addressed the concerns raised by the reviewers:

**1- Novelty and limitations of existing approaches:** To address concerns regarding novelty, we have explicitly highlighted the novelty of our approach in Section 1 (Lines 089 - 103) and the limitations of existing works in Section 2 ( Lines 153 - 169).

**2- Computational complexity and limitations:** In response to concerns about computational complexity and the limitations of our approach, we have included a discussion section before the conclusion (Page 10, Lines 502 - 525). This section provides a detailed analysis of the computational costs and possible areas for future improvement.

**3- Related work:** We have updated the related work section by removing unnecessary references to general online continual learning (OCL) methods (Page 3, Lines 121- 133) and focusing on studies on the classification of physiological signals in OCL settings (Page 3, Lines 135 - 161).

---

### Meta-Review · Area_Chair_XgQE · 2024-12-16

**Metareview:**

This paper addresses an method OPWA, which is an application of online continual learning to physiological data, tackling cross-subject variability. The main idea is in a prototype aggregation which synthesizes global prototypes from cross-subject prototypes. The paper is well written and the topic on handling subject variability is important in physiological data analysis. However, there are a few critical concerns raised by reviewers. In this paper continual learning is formulated subject-wise, instead of task-wise, which is not usual. One reviewer pointed out that subject-wise formulation is rather close to domain adaptation and generalization. There are lots of work on domain adaptation and generalization, which should be taken into account for future submissions. Cross-subject prototype weighting is a common technique, which limits the technical novelty of this paper since the contribution of this paper lies more in the combination of existing ones rather than in any novel algorithmic development.  Experiments should be also improved as well.  Therefore, the paper is not recommended for acceptance in its current form. I hope authors found the review comments informative and can improve their paper by addressing these carefully in future submissions.

**Additional Comments On Reviewer Discussion:**

During the rebuttal period, there was no change.  Most of reviewers stood by their original decisions.

---

### Decision · Program_Chairs · 2025-01-22

Reject